# Import of Non-Coding RNAs into Human Mitochondria: A Critical Review and Emerging Approaches

**DOI:** 10.3390/cells8030286

**Published:** 2019-03-26

**Authors:** Damien Jeandard, Anna Smirnova, Ivan Tarassov, Eric Barrey, Alexandre Smirnov, Nina Entelis

**Affiliations:** 1UMR 7156 GMGM Strasbourg University/CNRS, 67000 Strasbourg, France; damien.jeandard@etu.unistra.fr (D.J.); a.smirnova@unistra.fr (A.S.); i.tarassov@unistra.fr (I.T.); 2GABI-UMR1313, INRA, AgroParisTech, Université Paris-Saclay, 78350 Jouy-en-Josas, France; eric.barrey@inra.fr

**Keywords:** mitochondria, RNA import, PNPase, RNA importome landscaping, microscopy

## Abstract

Mitochondria harbor their own genetic system, yet critically depend on the import of a number of nuclear-encoded macromolecules to ensure their expression. In all eukaryotes, selected non-coding RNAs produced from the nuclear genome are partially redirected into the mitochondria, where they participate in gene expression. Therefore, the mitochondrial RNome represents an intricate mixture of the intrinsic transcriptome and the extrinsic RNA importome. In this review, we summarize and critically analyze data on the nuclear-encoded transcripts detected in human mitochondria and outline the proposed molecular mechanisms of their mitochondrial import. Special attention is given to the various experimental approaches used to study the mitochondrial RNome, including some recently developed genome-wide and in situ techniques.

## 1. Introduction

Mitochondria possess their own genome (mtDNA), which, in humans, encodes 11 mRNAs, 2 ribosomal, and 22 transfer RNAs required for the synthesis of 13 proteins of the oxidative phosphorylation complexes (Figure 1). However, this is far from being sufficient to perform all their functions, which necessitate more than 1000 proteins encoded by the nuclear DNA, synthetized in the cytosol and imported into mitochondria [1]. These include structural components, enzymes, and all the protein factors required for the maintenance and expression of the small mitochondrial genome. Additionally, in all groups of eukaryotes, some non-coding RNAs have been predicted or experimentally demonstrated to translocate into the mitochondria; this import pathway is often essential for the mitochondrial function [2,3,4]. In many cases, this is to be expected: the absence of an RNA gene whose product is strictly required for mitochondrial translation from the mitochondrial genome is usually considered a strong indication for the existence of a compensatory mitochondrial RNA import pathway. Indeed, in some species, tRNAs for select amino acids are not encoded in mtDNA and must be imported from the cytosol, ranging from just a few tRNAs in plants to the complete tRNA set in such protists as *Trypanosoma brucei* [5] and *Leishmania tarentolae* [6]. However, even in species in which all required tRNAs are encoded by the mitochondrial genome, tRNA import from the cytosol has been observed and is likely to become essential under particular stress conditions. This is notably the case of tRNA^Lys^_CUU_ (tRK1) in baker yeast *S. cerevisiae* [7,8]. The partial import of other cytosolic non-coding RNAs (including miRNAs and lncRNAs) has beens similarly reported in mammals, albeit their role inside the mitochondrial compartment is far from being obvious [9,10,11]. Such cases of seemingly superfluous targeting of nuclear-encoded RNAs into mitochondria beg the question of their functional significance and the mechanisms of their transport into the organelles. On the other hand, much of what we know about the RNA import phenomenon critically depends on the robustness of the analysis techniques employed to discover and characterize imported RNAs. Recent methodological developments relying on ingenuously tailored in situ and interactomic approaches promise to shed light on many of these questions.

In this review, we summarize and critically analyze the existing data on the RNA species imported into the human mitochondria (Table 1); the molecular mechanisms of the RNA targeting; and, in particular, the role of polynucleotide phosphorylase (PNPase) in this pathway. We also review classical and new methods to probe for the mitochondrial localization of RNAs and to comprehensively profile the small yet complex mitochondrial RNome.

## 2. Mitochondrial RNA Importome

The nuclear-encoded transcripts so-far detected in the human mitochondria are extremely diverse in size, structure, and primary function. However, all of them are non-coding RNAs, suggesting that the nucleus normally does not supply the mitochondrial genetic system with messenger RNAs, and all the polypeptides translated on the mitochondrial ribosomes are of strictly mitochondrial origin [11]. This situation actually seems to be common to all mitochondria and plastids, limiting the RNA import pathway to a bunch of usually small non-coding RNA species [2,3,4].

### 2.1. tRNAs

Human mtDNA encodes 22 tRNAs, which represents, taking into account the peculiarities of the mammalian mitochondrial genetic code, the minimal tRNA set sufficient for the translation of all mitochondrial mRNAs [35]. Surprisingly, however, some fully processed and apparently modified nuclear-encoded tRNAs have been detected inside the organelles by deep sequencing of RNA isolated from purified, RNase-treated mitoplasts (mitochondria devoid of the outer membrane) [11]. Some of them (e.g., tRNA^Leu^_UAA_ and tRNA^Gln^_UUG_) were enriched in comparison to crude mitochondria, further supporting their identification as potentially imported species. Although the authors have not provided a complete list of the enriched tRNAs, this study rose the questions of whether mammalian mitochondria can eventually obtain some tRNAs from the cytosol and whether those could be involved in mitochondrial translation.

Indeed, tRNA^Gln^_CUG_ and tRNA^Gln^_UUG_ were detected inside both rat and human mitochondria by RT-PCR, and the generic ability of isolated human mitochondria to internalize extraneous tRNA molecules of diverse origin in vitro has been demonstrated [9,10]. It was hypothesized that the import of cytosolic tRNA^Gln^ could support efficient mitochondrial translation in the case where the wobble position of the mitochondrial tRNA^Gln^ is not correctly modified and decoding of GAG codons becomes compromised, a mechanism similar to the one described in yeast [8]. However, such conditions and their corresponding adaptive mechanism in humans, if any, remain to be identified. Moreover, unlike the cytosolic compartment, where Gln-tRNA^Gln^ is obtained via direct glutaminylation by glutamine-tRNA synthetase (QARS), mitochondria use the indirect transamidation pathway, where tRNA^Gln^ is first misaminoacylated by the mitochondrial glutamyl-tRNA synthetase (mtEARS) and then converted into Gln-tRNA^Gln^ by the GatCAB complex [36]. Taking into account that human mitochondria lack QARS, and the GatCAB activity is required for mitochondrial translation [37], while the cytosolic tRNA^Gln^ is a poor substrated for GatCAB [36], it seems unlikely that imported tRNA^Gln^ may significantly contribute to the mitochondrial protein synthesis.

The ability of human mitochondria to import heterologous tRNAs in vivo has also been extensively studied. For instance, the yeast tRNA^Lys^_CUU_ (tRK1), one of the two nuclear-encoded tRNAs^Lys^, is partially redirected to mitochondria [7]. Although yeast mitochondria already encode a tRNA^Lys^ (tRK3) capable to decode both lysine codons, the wobble uridine of the tRK3 anticodon has a 2-thio modification, which cannot be efficiently formed at 37 °C, leading to a codon recognition defect under heat shock conditions. Thus, the import of tRK1, although dispensable at 30 °C, becomes essential for the translation of AAG codons at 37 °C [8]. Interestingly, when artificially expressed in human cells, tRK1 was similarly found to be imported into mitochondria [38]. Moreover, several tRK1-inspired transfer RNAs with altered amino acid identities retained the ability to go into the human mitochondria and could rescue mitochondrial mutations in cognate tRNA genes [38,39,40]. While this model is clearly artificial, it shows that at least a cryptic tRNA import pathway may exist in human cells and be exploited for therapeutic purposes.

### 2.2. 5S rRNA

5S ribosomal RNA (5S rRNA) is a component of the large subunit of most ribosomes, absent only in some mitochondria [41]. Located in the central protuberance, it acts as a scaffold interconnecting several functionally important sites on the ribosome [42,43,44]. No 5S rRNA homologue is encoded in the mammalian mitochondrial genome, but the nuclear-encoded 5S rRNA has been found in abundance in mammalian mitochondria by several teams [11,12,13,14,15]. For a long time, it was believed that 5S rRNA was incorporated in the mitochondrial ribosome. However, recent studies showed that mammalian mitochondrial ribosomes do not contain stably integrated 5S rRNA and instead harbor a mitochondria-encoded tRNA [45,46,47], leaving the mitochondrial function of 5S rRNA open to question. Interestingly, the knockdown of any of the three protein factors required for the 5S rRNA import in humans has led to a decrease of the mitochondrial translation and compromised the ability of cells to grow in the absence of glucose [19,48,49], an effect that cannot be always explained by other known functions of these proteins. This hints at a so-far unknown function of 5S rRNA in the mitochondrial protein synthesis that may involve the regulation of mitochondrial translation or the assembly of mitochondrial ribosomes.

### 2.3. RNase P RNA Component (H1 RNA)

RNase P is found in nearly all living organisms where its main function is to remove the 5′ leader sequence from tRNA precursors. This activity is critical for the processing of mitochondrial polycistronic transcripts according to the tRNA punctuation model [50,51]. Two types of RNase P have been found in different organisms: RNase P ribonucleoproteins, composed of a catalytic RNA component (H1 RNA) associated with one or several proteins, and protein-only RNases P (PRORP) [52,53]. For example, nuclear RNase P in humans and yeast is a ribonucleoprotein, a situation typical for opistokonts. On the contrary, although in some species mitochondrial RNase P contains both an RNA and protein subunits (and in many fungi the RNase P RNA component can even be encoded in mtDNA), in the majority of cases, mitochondria rely on nuclear-encoded PRORP enzymes to process their tRNAs [53]. This is also the case in the human mitochondria, where the RNase P activity is provided by a well-studied protein-only complex composed of three subunits [54]. Nevertheless, a ~340-nt transcript corresponding to the nuclear H1 RNA has also been detected in mitochondria and suggested to participate in mitochondrial tRNA processing [11,17,18,19]. Although some bacteria and archaea, indeed, have both a RNA-based and a protein-only RNases P [55], such a situation appears to be unique among mitochondria. Of note, to date there is no evidence that a functional RNase P ribonucleoprotein can be assembled inside human mitochondria (since the H1 RNA-binding proteins are normally localized in the nucleus), arguing against the role of the imported H1 RNA in mitochondrial RNA processing.

### 2.4. RNase MRP RNA Component (RMRP)

In the nucleus, the catalytic RNA component of mitochondrial RNA processing (MRP) ribonuclease (also known as RMRP), which is evolutionarily related to the H1 RNA [56], is involved in the 5′ end maturation of 5.8S rRNA [57]. RMRP has also been reported to play a role in the cell cycle regulation [58] and to associate with the telomerase reverse transcriptase (TERT) for the synthesis of double-stranded RNA further processed into short interfering RNAs [59]. While RMRP mainly localizes in the nucleus, a small part of its pool has been found in mammalian mitochondria [22]. Its possible function was initially studied in isolated mouse mitochondria, where it was supposed shown to process the RNA primer for the replication of the mtDNA heavy strand [20,60]. However, recent studies have shown that the RNA primer formation is a consequence of a premature arrest of the mitochondrial RNA polymerase after a G-quadruplex situated in the control region of human mtDNA, which takes place when RNA polymerase is not associated with the transcription elongation factor TEFM [61]. Moreover, it was demonstrated that only the 3′ half (~130 nt) of RMRP can be found inside mitochondria, the RNA being processed upon or during the mitochondrial import [20,21]. Given the specific pseudoknotted structure of RMRP, which is required for its function, this cleavage would normally result in a loss of catalytic activity [56]. Therefore, it is unlikely that RMRP acts as a nuclease in the mitochondrial compartment. However, a recent study has shown that human RMRP interacts with GRSF1, an important component of mitochondrial RNA granules [62,63], suggesting that it may still be involved in mitochondrial RNA metabolism.

### 2.5. SAMMSON

SAMMSON is a lncRNA that is predominantly expressed in aggressive melanomas, where it is required to stimulate the mitochondrial function of actively proliferating tumor cells [23,24]. SAMMSON localizes in the cytoplasm and is proposed to interact with the protein p32, another pro-oncogenic factor involved in the regulation of mitochondrial gene expression [64,65]. Knockdown of SAMMSON impairs the p32 targeting to the mitochondria, whose activity is essential for quickly dividing melanoma cells and results in a mitochondrial protein synthesis defect. Because SAMMSON was found to colocalize and copurify with mitochondria, it was suggested that its involvement in p32 traffic could be accompanied by its own mitochondrial import [23], which still needs to be confirmed with alternative approaches. Importantly, both SAMMSON and its role in p32 transport are likely specific to melanoma cells, since in a variety of other models p32 does not seem to require special factors for efficient import into the mitochondria [66,67,68].

### 2.6. hTERC

Recently, the RNA component of the human telomerase, hTERC, encoding the sequence of the simple repeats added to the ends of DNA (telomere) [69], has also been proposed to be imported into mitochondria [25]. In this study, the import of hTERC was initially suspected based on the presence of a region similar to a short stem-loop that, in H1 RNA and RMRP, facilitates the mitochondrial localization of RNAs (see Figure 3c below) [19]. Indeed, the presence of hTERC in purified mitoplasts treated with micrococcal nuclease was confirmed by RT-PCR. Surprisingly, deletion of this sequence from hTERC significantly increased its mitochondrial import instead of abolishing it. Similarly to RMRP, hTERC was found to be processed upon mitochondrial import into a shorter 195 nt-long RNA termed TERC-53. Since the processed version of hTERC was detected mostly in the cytosol, the authors suggested that TERC-53 is re-exported from the mitochondria, which would permit to somehow relay the functional state of the mitochondria to the nucleus and other cellular compartments [25,70]. However, evidence that hTERC processing occurs within the mitochondrial matrix and not at the mitochondrial surface is currently lacking.

### 2.7. microRNAs

miRNAs are non-coding RNA molecules of about 22 nucleotides that target and repress mRNAs at a level of stability and/or translation through the recruitment of the RNA-induced silencing complex (RISC) [71,72]. They are transcribed as primary microRNAs (pri-miRNAs), which are processed in the nucleus by the DROSHA/DGCR8 complex into precursor (pre)microRNAs, exported to the cytoplasm and further cleaved by DICER1 into mature miRNAs. Then, they associate with the RNA-binding protein Argonaute 2 (AGO2), a component of RISC, and bring the complex to the target mRNA, usually by annealing to a specific region in the 3′ UTR.

Pioneered with RT-qPCR- and microscopy-based analyses and more recently powered by deep sequencing, the discovery of miRNAs localized in mammalian mitochondria has soared over the last decade [11,27,28,31,32,33]. While most mitochondrial miRNAs seem to be nuclear-encoded and imported inside the mitochondria or associated with the outer mitochondrial membrane, a few miRNAs have been suggested to be produced by mtDNA itself [27,28,31,73,74,75]. An in silico analysis has revealed multiple putative miRNA-binding sites on the mitochondrial DNA [27]. However, the possible mode of action of microRNAs in the mitochondrial compartment is unclear, since mitochondrial mRNAs contain no or small 3′ UTRs, and only one component of RISC—AGO2—has been proposed to localize to the mitochondrial matrix [26,30,31]. Indeed, some mitochondria-localized miRNAs have been found to enhance the synthesis of select mitochondrial proteins instead of inhibiting it. For example, miR-1 is able to stimulate the translation of *MT-ND1* and *MT-CO1* mRNAs in an AGO2-dependent manner, while repressing some nuclear DNA-encoded mRNA targets in the cytosol, both pathways being important for muscle differentiation in mouse [30]. On the contrary, miR-181c has been suggested to repress the expression of *MT-CO1* in rat heart by targeting its 3′ UTR [29,76], and miR-378 has been found to downregulate *MT-ATP6* in mouse HL-1 cells [77]. Other non-canonical functions of mitochondrial miRNAs can be expected, and many hypotheses have been proposed, including targeting mitochondrial transcription in human [34]. However, a convincing demonstration of the corresponding molecular mechanisms is notoriously difficult, given that miRNAs may affect the expression of nuclear genes involved in mitochondrial biogenesis and function, and ruling out such indirect effects caused by their canonical action in the cytosol is anything but easy.

Besides its role in regulation of mitochondrial gene expression, it has been suggested that imported miRNAs may undergo post-transcriptional modification in the mitochondria or on their surface, leading to structural changes and adjustment in their RNA and/or protein interaction specificities, before being released back to the cytosol [26,78]. It is also important to keep in mind that the sets of mitochondria-associated miRNAs found in various cell types show a remarkably poor overlap: in human, only seven miRNAs have been identified in at least three different studies, and most mitochondrial miRNAs have only been reported in one or two studies [26]. Such low reproducibility, although partially imputable to cell type variations, raises urgent questions with regard to the methodology used to identify and validate potentially imported miRNA species.

## 3. Mechanisms of RNA Import

### 3.1. RNA Targeting to the Mitochondria in Various Eukaryotes

Although the mitochondrial RNA import is ubiquitously found among eukaryotes, the mechanisms that direct mitochondrial targeting and transfer of RNAs into the mitochondrial matrix seem to differ greatly between organisms (Figure 2) [2,79,80]. In general, mitochondrial RNA import involves three criteria: (1) the presence of selective signals, or import determinants, within the imported RNA; (2) a mechanism to intercept RNA from its cytosolic location and redirect it to the mitochondrial surface; and (3) a translocation pathway to transfer the RNA molecule across the mitochondrial envelope.

Mechanisms of RNA import into plant mitochondria remain elusive (Figure 2a). Generally, only a few cytosolic tRNAs translocate into the mitochondria to replace their missing counterparts. In vitro import tests have suggested that the tRNA import did not require cytosolic factors; however, mutations disrupting aminoacylation of tRNAs prevented their import in vivo, and addition of protein factors improved tRNA import rates in vitro. This indicates the potential involvement of aminoacyl-tRNA synthetases as carrier proteins to direct tRNA to plant mitochondria [79,81], although other unidentified factors may also participate in this process. The mitochondrial tRNA transport across the membranes is ATP-dependent and requires the presence of the membrane potential [82]. VDAC (porin) has been identified as the main channel for translocation of tRNAs across the outer mitochondrial membrane in potato, with the participation of complexes of the protein import machinery (Tom20 and Tom40) [83,84]. However, the mechanism of translocation through the inner membrane remains unknown.

The pervasive mitochondrial tRNA targeting in the kinetoplastid *T. brucei* (Figure 2b) seems to be specified by a signal in the T-loop of tRNAs recognized by the cytosolic elongation factor eEF1α, thus only excluding tRNA^Met^_i_, which does not bind eEF1α, from this pathway [85]. The translocation into the mitochondrial matrix is also ATP-dependent and occurs through the protein import apparatus ATOM (archaic translocase of the outer membrane) complex [86,87].

### 3.2. Mitochondrial RNA Targeting in Yeast and Human

Interestingly, the yeast and human RNA import pathways seem to share several features (Figure 2c). In both cases, interplay of specific mitochondria-related cytosolic proteins is required to intercept the imported RNA from its primary location and redirect it toward the mitochondria, followed by translocation, which at least partially depends on the pre-protein import apparatus. In yeast, the glycolytic enzyme enolase (Eno2p), probably helped by some additional cytosolic factors, first interacts with tRK1 and brings it to the mitochondrial surface [88]. Interaction of Eno2p with tRK1 induces a conformational change of the tRNA so that the 3′ extremity of the tRNA folds back on the 3′ side of the T-arm to form the so-called F-stem-loop (Figure 3a) [89,90,91]. This rearrangement also involves the anticodon arm and the variable region, while the D-arm remains intact. This alternative tRK1 structure has an improved affinity to the second protein factor, precursor of the mitochondrial lysyl-tRNA synthetase (preMSK), which mediates the tRK1 transport to the mitochondrial matrix. Interestingly, the F- and D-stem-loops grafted on heterologous transcripts were sufficient to induce mitochondrial import of small synthetic RNAs in vivo [90]. Of note, since both enolase and the mitochondrial lysyl-tRNA synthetase are also present in human cells, they could sustain the import of exogenously introduced tRK1 into human mitochondria [92,93]. Import of tRK1 and its synthetic derivatives bearing the F- and D-stem-loops have been used to mitigate the consequences of some pathogenic mutations in mtDNA [39,94].

The 5S rRNA import in human mitochondria follows a similar logic (Figure 2c) [95]. 5S rRNA is transcribed by RNA polymerase III with the help of the transcription factor IIIA (TFIIIA) and is exported to the cytosol in complex with TFIIIA, where it interacts, through its β-domain, with the cytosolic ribosomal protein L5 (Figure 3b) [96]. 5S rRNA is then imported back to the nucleus to be incorporated in nascent cytosolic ribosomes. In the cytosol, 5S rRNA can alternatively interact, through its γ-domain, with the precursor of the mitochondrial ribosomal protein L18 (MRP-L18, or its recently described cytosolic isoform [97]) and be redirected to the mitochondria [49]. Similarly to tRK1 and enolase, the interaction with MRP-L18 induces a change in the conformation of the 5S rRNA molecule and allows for the interaction of its α-domain with the second protein factor, rhodanese, and subsequent import of the RNA-protein complex into the mitochondrial matrix [15,48]. Both α- and γ-domains are essential for mitochondrial targeting of 5S rRNA, whereas the β-domain is only involved in 5S rRNA incorporation in the cytosolic ribosome. Not surprisingly, the removal of the β-domain or its substitution with a short RNA sequence does not alter the import of 5S rRNA and may even increase its targeting to mitochondria [95].

The mechanisms of mitochondrial targeting of other RNAs are much less studied but may similarly involve some protein factors destined for the mitochondrial matrix. For example, a recent study showed that, in humans, the RMRP export from the nucleus is mediated by the protein HuR and, once in the mitochondria, this RNA interacts with GRSF1 [21]. One of the two proteins may also be responsible for the targeting of RMRP to the mitochondrial surface. Similarly, the main candidate for the mitochondrial targeting of mammalian miRNAs remains AGO2, also detected inside the organelles [30]. Finally, one cannot exclude that some protein factors required for mRNA localization on the mitochondrial surface [98,99] may be recruited to bring along select non-coding RNAs. Further studies are required to verify these hypotheses.

### 3.3. Translocation through the Mitochondrial Membranes and the Role of Pnpase

Mechanisms of RNA translocation across the mitochondrial membranes in humans are not well understood. It has been shown that the process is ATP-dependent, requires the presence of the membrane potential, and might involve the protein import apparatus, as has been observed in yeast [14,100]. The RNA translocation through the outer mitochondrial membrane (OMM) may be mediated by the TOM complex and/or VDAC channel (our unpublished results).

Much more is known about the possible mechanism of RNA translocation through the intermembrane space (IMS) involving polynucleotide phosphorylase (PNPase, Figure 2c and Figure 4) [19]. PNPase is an evolutionarily conserved 3′-to-5′ exoribonuclease encoded in human by the *PNPT1* gene. It possesses an N-terminal signal for mitochondrial localization and has been detected in both the mitochondrial matrix and IMS [101,102]. In the matrix, PNPase, in complex with the DNA/RNA helicase SUV3, is involved in general RNA decay, *MT-ND6* mRNA processing, non-coding RNA clearance, and R-loop resolution [103,104,105,106,107]. In IMS, it is believed to be a factor in RNA import in the mitochondria [19], to control the escape of mitochondrial double-stranded RNA into the cytosol under physiological and disease conditions [108], and may, upon release into the cytosol, also play a role in apoptosis [109]. Mutations in human PNPase are linked to mitochondrial diseases of varying severities, characterized by respiratory chain deficiencies and associated with both mitochondrial RNA processing defects and perturbed RNA import [110,111,112,113]. Given the pervasive impact of PNPase on the mitochondrial RNA metabolism, it is not always possible to disentangle the observed phenotypes and conclusively connect them with various functions proposed for this protein.

Since PNPase is a nuclease, the mechanism by which it could be involved in RNA import remains puzzling, although bacterial PNPase homologues are known to eventually act as simple RNA-binding proteins without degrading their associated transcripts (reviewed in [114]). It was postulated that PNPase can recognize stem-loop structures [19], which correspond to the import determinants identified in 5S rRNA, tRK1, H1 RNA, and RMRP (Figure 3). Structural studies permitted to propose a mechanism by which human PNPase selects RNAs for mitochondrial import [115]. A PNPase monomer consists of two RNase PH domains (PHI and PHII); a so-called α-helical domain; and two RNA-binding domains, KH and S1 (Figure 4a). Three monomers assemble into a “doughnut”-shaped homotrimer with a central channel formed by the six RNase PH domains, where the catalytic site is located. The KH domains were shown to be important for the single-stranded RNA binding. They form a “pore” at the entry of the PH channel through which the 3′ end of a single-stranded RNA enters and subsequently interacts with the catalytic site (Figure 4b) [115]. However, the pore is too narrow to allow the passage of a structured RNA. A more recent study has proposed that the S1 domains, also forming a pore atop of the KH domains, interact with structured double-stranded RNAs [113]. If the length of the 3′ overhang of the bound RNA is too short, it will prevent the entry of the RNA into the KH pore and the PH channel, protecting it from degradation (Figure 4b) [19,115]. In future studies, ideally involving mutations that permit to dissociate the ribonuclease and RNA-binding functions of PNPase would shed light on its actual role in the RNA import pathway and set the stage for the long-awaited functional analyses of the imported transcripts in the mitochondrial compartment.

## 4. Identification and Validation of Imported RNAs

Besides the RNA species described above, many other types of small non-coding RNA have been detected in human mitochondria, including snRNAs, snoRNAs, piRNAs, and tRNA fragments [11,28]. To verify whether those small RNAs are indeed located inside the mitochondria or represent mere cytosolic contaminants, adequate experimental techniques are essential, and diverse strategies have been proposed to this end. They correspond to five conceptually different methodologies: (1) subcellular fractionation and analysis of RNA isolated from RNase-treated mitochondria or mitoplasts, (2) in vitro import into isolated mitochondria, (3) microscopy-based approaches, (4) compartment-specific tagging techniques, and (5) compartment-specific crosslinking methods.

### 4.1. Classical Cellular Fractionation

Identification of RNA in isolated mitochondria has been the first approach used to discover imported RNAs (Figure 5). It relies on isolation of highly pure mitochondria and typically involves RNase treatment, which is expected to selectively degrade contaminant RNAs sticking to the mitochondrial surface while sparing the transcripts protected by the inner mitochondrial membrane in the matrix [116,117].

#### 4.1.1. Isolation and Purification of Mitochondria

Isolation of mitochondria requires an initial step of cell homogenization in conditions that maintain mitochondrial integrity, usually followed by differential centrifugations [118]. Low-speed centrifugation allows for the removal of cellular debris and nuclei while mitochondria remain in the supernatant. They are then recovered by high-speed centrifugation and typically subjected to additional purification steps like centrifugation in Percoll gradients or through sucrose cushions to eliminate light membrane-bounded organelles [119]. Alternatively, mitochondria can be recovered from disrupted cells with the use of anti-TOM22 antibody-coated magnetic beads [27,31,120], which is faster and associated with less contamination with cosedimenting organelles, albeit at a cost of a more modest yield.

In either case, the isolated crude mitochondria remain contaminated with other cellular components such as the endoplasmic reticulum [121]; Golgi; endosomes; and RNA-containing organelles such as P-bodies [78], stress granules [122,123], and cytosolic ribosomes engaged in translation at the surface of OMM [124,125]. To improve the purity of the mitochondrial sample, the outer membrane is usually removed through the generation of mitoplasts. This can be achieved by two different approaches: either with detergents such as digitonin, which preferentially permeabilizes the cholesterol-rich OMM, or by inducing slight swelling of mitochondria and rupture of the outer mitochondrial membrane by osmotic shock [126,127]. Since the conditions of treatment are always a tradeoff between the removal of OMM and the preservation of the integrity of IMM [128], moderate treatments are preferred, albeit they can only partially destroy the outer membrane in a subpopulation of mitochondria [127,129]. Mitoplasts are also usually treated with ribonucleases, unable to cross IMM, to digest remaining cytosolic RNA contaminants. RNA extraction can then be performed to analyze the mitochondrial RNA composition.

This RNase treatment step is critical as the choice of RNase, and the conditions of digestion have a direct and decisive impact on the interpretation of the results of the analysis. For instance, different enzymes have varying specificities and unequal abilities to degrade RNAs, and some types of transcripts (e.g., miRNAs) are significantly more resistant to RNase-mediated degradation [130] or extensively protected by associated proteins, making RNase-based methods of little value in assessing their mitochondrial localization. Identification of such false-positive cases requires special controls involving the rupture of the mitochondrial membranes with a detergent to prove that the observed protection was due to the membranes and not to other confounding factors.

#### 4.1.2. Analysis of the Samples

In a hypothesis-driven analysis, when the identities of candidate RNAs are known, northern blotting and RT-qPCR can be performed to evaluate their presence in the isolated mitoplasts. This is normally accompanied by quantification of several negative controls (bona fide cytosolic contaminant RNAs) that should be absent or strongly depleted from the mitoplast preparation.

However, to analyze the mitochondrial RNAs in an unbiased and high-throughput way and detect potentially new imported transcripts, next-generation sequencing should be opted for. The experimental design of RNA-seq and the post-analysis pipeline vary greatly in function of the model system and the biological question. The sequencing of RNA itself may exploit different methodologies, depending on the library generation protocol and the commercial instrument used [131]. The diversity of RNA-seq approaches, including size selection steps, facultative rRNA depletion, differences in equipment, reagents, depth, and sensitivity, may significantly complicate the comparison of data obtained by different laboratories.

Large-scale analyses of mammalian mitochondrial RNA samples by RNA-seq have been attempted by many laboratories [11,28,31]. However, in most cases it has remained unclear to what extent the observed nuclear-encoded RNAs could be imputed to cytosolic contamination. Indeed, at sufficient depth, RNA-seq provides high detection sensitivity capable to reveal trace amounts of contaminant RNA, and it appears practically impossible to obtain completely pure mitochondria by conventional isolation methods. Consequently, the mitochondrial RNome studies published so-far have been successful in charting the mitochondrial transcriptome [11] but have often been indecisive and sometimes even contradictory concerning the nuclear-encoded RNAs residing in the organelles.

Obviously, the problem of robust identification of the imported RNAs with RNA-seq cannot be solved in a naïve “simple detection” mode. Instead, better-controlled enrichment-based approaches should be adopted. For instance, comparison of transcript levels between mitoplast- and crude mitochondria-derived RNA samples offer the first simple means to prioritize potentially imported RNA candidates and dismiss some typical abundant contaminants [11]. Since RNA-seq remains the only convenient strategy for large-scale analysis of RNomes, further development of robust enrichment measures and informative controls will be required in order to draw an accurate list of mitochondrially imported RNAs. It has also become increasingly obvious that the results generated by genome-wide approaches need to be validated with orthogonal techniques.

### 4.2. In Vitro RNA Import into Isolated Mitochondria

A simple and popular way to verify the ability of RNA to penetrate into the mitochondria and at the same time study the biochemical requirements for this process is the in vitro import test (Figure 5) [116,117]. For this, a radiolabeled RNA transcript of interest is added to isolated crude mitochondria, eventually in the presence of ATP, ATP-regeneration system and cytosolic protein factors. Upon incubation, a ribonuclease treatment is performed to degrade the RNAs outside the mitochondria, whereas the RNAs in the mitochondrial matrix are protected from degradation by the mitochondrial membranes. Isolated RNAs are then separated by polyacrylamide gel electrophoresis, and the presence of the candidate RNA is assessed by autoradiography. It is important to keep in mind that this approach may only provide proof for the intrinsic ability of RNA to go into the mitochondria, without guaranteeing that this actually happens in vivo. Therefore, this method has to be combined with alternative in vivo or in situ approaches.

### 4.3. Imaging of Mitochondrial RNAs in Intact Cells

Strong evidence of the specific localization of an RNA in mitochondria can be obtained by its direct observation in cells by fluorescence microscopy. Various approaches have been developed to this end. The first group of methods is based on the introduction of exogenous RNAs tagged with fluorophores, fluorogenic aptamers, or protein-binding domains. This can be achieved either by direct transfection of cells with RNA or by its expression from a plasmid or a stably integrated recombinant gene [132,133]. An immediate advantage of this technique is the possibility to track the motion of recombinant RNA molecules in live cells. Another group of methods is based on the detection of endogenous RNAs and does not need introduction of recombinant transcripts. In live cells, endogenous RNAs can be imaged with the use of designer sequence-specific Pumilio homology domain mutants [134]. Nowadays, the most popular and reliable technique for direct observation of endogenous RNAs is fluorescence in situ hybridization (FISH). However, it requires cell fixation, which is incompatible with live cell imaging.

#### 4.3.1. Recombinant RNA Probes

Contrary to some proteins, RNAs cannot produce intrinsic fluorescent signal. Thus, fluorescence tools that had already been successfully used to study cellular proteins, such as the green fluorescent protein (GFP), had to be adapted to image RNAs in cells. MS2 tagging was the first system to provide GFP-mediated imaging of RNA localization [135]. The bacteriophage MS2 coat protein, which can be fused to GFP, is specifically recruited to a short RNA element consisting of two hairpins (MS2 aptamer) that can be inserted into RNA of interest. However, unbound GFP results in high background that can mask specific fluorescent signal. In order to increase the signal-to-noise ratio, multiple repeats of the MS2 aptamer must be incorporated in RNA, leading to long insertions of hundreds of nucleotides and potentially affecting the behavior of the tagged transcript [136].

Improvement of the technique was achieved with the development of a series of RNA aptamers that can capture specific fluorophores and become fluorescent. Unbound fluorophores are poorly fluorescent and light up only when bound by an RNA aptamer [137,138]. Several aptamers were successfully developed, including spinach, broccoli, and mango [15,139,140], with each next generation providing a better affinity for fluorophore, higher fluorescence, and better resistance to photo-bleaching. The latest improvement in the RNA aptamer-based fluorescent system has allowed for the direct detection of 5S rRNA in human cells and confirmed its γ-domain-dependent colocalization with mitochondria [15]. It is important to note that, even though RNA aptamers are relatively small (40 to 100 nt), their presence modifies the structure of the concerned RNA, leading to a potential loss of RNA function or proper localization, which is particularly relevant for short RNAs. Therefore, detection of endogenous RNAs looks especially appealing as it permits one to avoid disruption of the RNA structure and transcript mislocalization.

#### 4.3.2. Imaging of Endogenous RNAs

An original protein-based probe for imaging of endogenous RNAs has been developed on the basis of Pumilio homology domain (PUM-HD) mutants that have the ability to recognize single-stranded RNAs in a programmable and modular manner [134]. Ozawa et al. constructed two mitochondrially targeted Pumilio proteins directed to two eight-base sequences in the mitochondrial *MT-ND6* mRNA, which reconstitute a full-size fluorescent protein upon RNA binding [141]. This approach permitted one to monitor the *ND6* mRNA dynamics in living mammalian cells. However, using the Pumilio system requires extensive molecular design, and potential formation of protein dimers or oligomers may produce false-positive signal.

In a simpler and more straightforward way, detection of specific nucleic acids in the cell can be performed by fluorescence in situ hybridization (FISH) [142]. In its simplest version, the RNA molecule of interest is hybridized with a sequence-specific oligonucleotide labeled with a fluorophore. This technique was employed to detect abundant human mitochondrial transcripts such as the *MT-ND6* mRNA and 12S rRNA [63]. The same principle was applied for the visualization of mitochondrial transcripts in a study addressing the spatio-temporal organization of transcription and replication in mitochondria [143]. Application of locked nucleic acid (LNA)-modified oligonucleotides made possible the detection of much less abundant and short microRNAs in human mitochondria [27].

However, the canonical variant of FISH has limited sensitivity since a probe labeled with only one fluorophore provides a weak signal-over-background ratio. To bypass this limitation and additionally make the FISH technique truly quantitative, the so-called single-molecule FISH has been introduced [144]. It is based on the hybridization of multiple fluorescently labeled oligonucleotides to visualize individual transcripts as diffraction-limited spots. Initially, this approach was almost exclusively used to detect long transcripts, like mRNAs and ribosomal RNAs, since it requires hybridization of 20 to 30 oligonucleotides with the length ~20 nt. This limitation has been overcome by combining FISH with an amplification system. For example, branched DNA technology (bDNA, Advanced Cell Diagnostics, and Affimetrix) increases the signal-to-background ratio using specific DNA probes annealing to a target RNA molecule and serving as a platform for subsequent binding of numerous fluorophore-labelled oligonucleotides [145]. This approach was applied for the detection of the 11 mitochondrial mRNAs in a high-throughput quantitative image-based transcriptomic study in human cells [146]. We found that this technology can be adapted for the detection and quantification of mitochondrially encoded rRNAs, low-abundance non-coding RNAs, and even short transcripts like tRNAs (Figure 6a).

Despite significant advances in RNA labeling technology, studies of mitochondrial RNA import by fluorescence microscopy remain a challenging task because of the dual (cytosolic and mitochondrial) localization of the imported molecules and generally low import efficiency. Furthermore, the very size of mitochondria borders the resolution limit of the conventional fluorescence microscopy [147]. Thus, it is impossible to clearly distinguish between RNAs located inside the mitochondria and those simply attached to the mitochondrial surface, as can be seen for the highly abundant 5S rRNA (dual localized) and 5.8S rRNA (strictly cytosolic) in human cells (Figure 6b). In recent years, due to a breakthrough in the super-resolution microscopy, it has become possible to achieve a resolution up to 10 times higher than that imposed by the diffraction limit, allowing for precise sub-organellar localization of mitochondrial proteins [148,149,150]. However, very few studies have so-far attempted to combine RNA smFISH and super-resolution microscopy. One example is an elegant study of the subcellular distribution and the base-pairing dynamics of an sRNA and an mRNA in *E. coli* [151]. In contrast, RNA imaging by diffraction-unlimited microscopy in human cells is still an uncharted area with unclear technical limitations. Super-resolution microscopy holds the promise to answer some nagging questions about the mitochondrial RNA import and glean important insight into the functions of nuclear-encoded RNAs within the mitochondria. However, there is still a big road to go until it will be possible to routinely use this tool in assessing the fine submitochondrial localization of the imported transcripts.

### 4.4. Mitochondria-Specific Tagging of RNAs

A conceptually different group of methods to assess the subcellular localisation of transcripts builds on in vivo chemical modification of RNAs confined in a membrane-bounded compartment, such as mitochondrion. This chemical “mark” permits one to specifically enrich and identify modified RNAs. In a recent embodiment of this principle (Figure 7a), it was proposed to exploit the guanosine oxidation with singlet oxygen generated by an irradiated fluorescent dye [152,153]. Singlet oxygen has very short lifetime and, therefore, small diffusion radius, making it an excellent reagent for proximity labeling. It easily oxidizes nearby guanosines to 8-oxoguanozines, which readily react with nucleophiles, e.g., propargyl amine. RNAs with guanosines derivatized with propargyl amine can be specifically labeled in a click reaction with azide-biotin and isolated and analyzed by RT-qPCR or another method. The compartment-specific localization of fluorescent dye is achieved through its stable association with a Halo-tag fusion to a protein with the required localization. For instance, the authors successfully and site-specifically tagged RNAs in the nucleoplasm or nucleoli of human cells [152,153]. It will be exciting to apply this method to profile mitochondria-localized RNAs, especially in combination with RNA-seq as detection technique.

The recent ClickIn approach exploits an in organello click reaction to verify the uptake of molecules, such as peptides and peptide nucleic acids, by isolated rat mitochondria [154]. Organelles pre-loaded with the mitochondria-targeted cyclooctyne MitoOct are incubated with an azide-labeled substrate (Figure 7b). If the molecule is internalized by the mitochondria, a click reaction takes place and the resulting adduct can be detected by mass spectrometry. Although this method is currently available only for isolated mitochondria, it may provide an excellent alternative to RNase-based approaches in assessing the mitochondrial import of short transcripts, such as miRNAs.

### 4.5. Spatially Restricted Crosslinking-Based Techniques

A methodologically similar group of approaches is based on compartment-specific crosslinking of RNAs to proteins for their subsequent isolation and identification. Two slightly different variations of this method have been published so-far: APEX-RIP [155] and Proximity-CLIP [156] (Figure 7c). In both cases, ascorbate peroxidase (APEX) is targeted to the subcellular location of interest where it catalyzes the oxidation of biotin-phenol by hydrogen peroxide. The resulting biotin-phenoxyl radicals are extremely short-lived and covalently modify nearby proteins in a radius of <5 nm. This enables their subsequent isolation on streptavidin beads, and identification by mass spectrometry, which has been successfully used to profile the human mitochondrial proteome with an exquisite spatial resolution permitting to distinguish matrix- and IMS-localized proteins [157]. To expand the scope of this method to RNAs, APEX-RIP introduced a formaldehyde crosslinking step, whereas Proximity-CLIP relies on UV crosslinking to covalently attach RNAs to neighboring proteins [155,156]. The RNAs co-purified with biotinylated proteins under stringent conditions and became significantly enriched in comparison to negative control IPs (e.g., without crosslinking or biotin-phenol or hydrogen peroxide), which are considered bona fide residents in the analyzed compartment. For instance, by using a mitochondrially targeted mito-APEX2 protein, the APEX-RIP technique has recently permitted one to profile mitochondrial RNAs in HEK293T cells [155]. Interestingly, the authors did not detect any long nuclear-encoded RNAs, while they readily identified all long mitochondria-encoded transcripts. The method, however, remains to be applied to small RNAs, as they constitute the majority of the RNAs identified or suggested to be imported into mitochondria.

## 5. Unsolved Questions and Future Directions

Although the history of research in mitochondrial RNA import counts now four decades, many nagging questions persist. How do the RNAs destined for mitochondria get selected from the cellular milieu and transported to the mitochondrial surface? Does this process differ between cell types and metabolic conditions? How do RNAs cross the inner mitochondrial membrane? Where are they located inside the organelles and, most importantly, what do they do there? With the recent development of microscopy and other in situ methods and interactomic and molecular biology approaches, it seems that the long-awaited methodology is coming of age to tackle these problems in a more conclusive way.

Our knowledge of the spatial organization of the RNA metabolism in human cells has experienced a qualitative breakthrough over the last few years [158,159]. Characteristic markers of various submitochondrial locations have been identified, and their associated functions are finally coming to light, meaning that unique insights into the role of mitochondria-localized transcripts could be gained from combined studies of their interactions with RNA-binding proteins [160], genetic approaches, and fine-scale microscopy analyses (e.g., expansion microscopy [161] and super-resolution techniques).

The question of which nuclear-encoded RNAs are imported in mammalian mitochondria is a matter of controversy, and some speculative arguments (e.g., inability of imported RNA species to perform their canonical functions in the mitochondrial compartment) have been forwarded [162]. Taking into account that alternative RNA functions are a widespread phenomenon (consider, for example, the multiple roles of tRNA [163]), such considerations certainly do not permit to rule out the existence of mitochondria-localized pools of select cytosolic RNAs. Moreover, one cannot exclude that many of the imported RNAs do not have any function (yet) and simply represent evolutionary material, which may, in distant future, contribute to a progressive replacement of lost mitochondrial genes, as happened in other eukaryotic lineages. We believe that only solid experimentation relying on a combination of several orthogonal approaches can provide a decisive answer to the question of which RNAs are delivered into the mitochondria and which are not. In this regard, recent in situ and in vivo techniques, including APEX-RIP and spatially restricted tagging approaches, which do not require subcellular fractionation and RNase treatment, are particularly promising.

## Figures and Tables

**Figure 1 cells-08-00286-f001:**
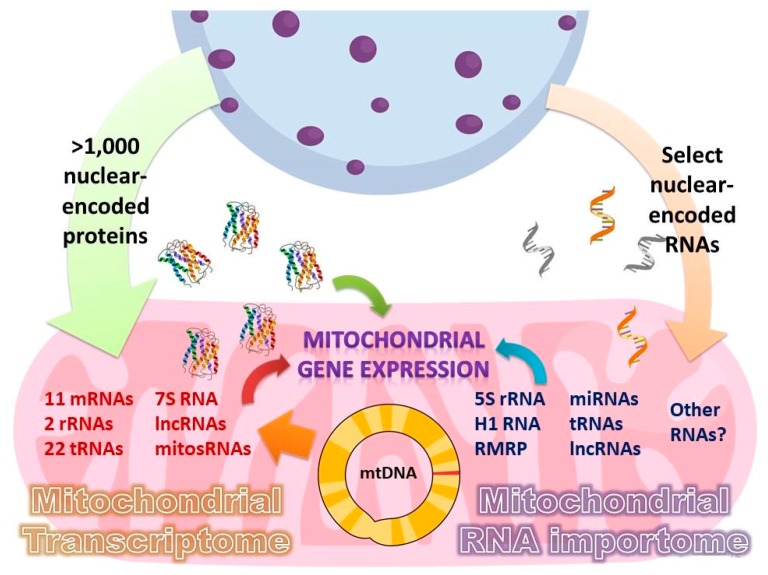
Human mitochondrial proteome and RNome are chimeric.

**Figure 2 cells-08-00286-f002:**
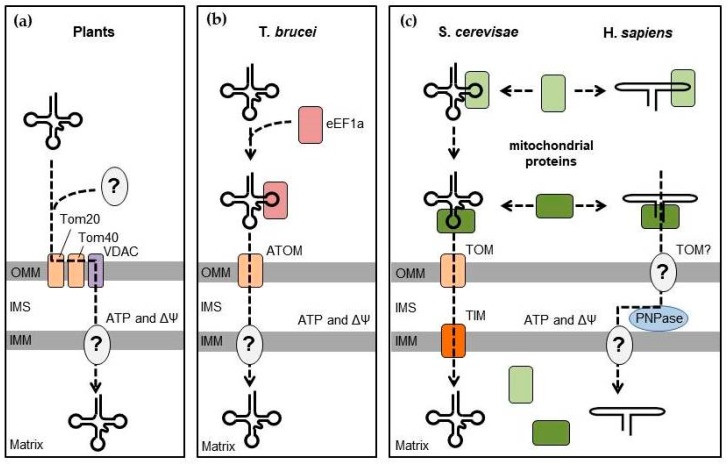
RNA import mechanisms in various eukaryotic models. (**a**) In plants, the targeting of RNA to mitochondria probably relies on some unidentified cytosolic factors (e.g., aminoacyl-tRNA synthetases). Translocation across the outer mitochondrial membrane (OMM) requires Tom20, Tom40, involved in the pre-protein import, as well as the voltage-dependent anion channel (VDAC). The mechanism of translocation across the inner mitochondrial membrane (IMM) remains unknown. (**b**) In *T. brucei*, tRNAs are directed to the mitochondrial import by the translation elongation factor eEF1α, while the archaic translocase of the outer membrane (ATOM), which also transports mitochondrial pre-proteins, mediates their translocation though OMM. (**c**) Mitochondrial RNA import in *S. cerevisiae* and humans. RNAs are targeted to mitochondria by various nuclear-encoded proteins normally localized inside or in the vicinity of mitochondria. In *S. cerevisiae*, translocation into the mitochondrial matrix is mediated by the protein import machinery (translocases of the outer and inner membranes, TOM, and TIM). In humans, the translocation mechanism is unknown but seems to require the protein PNPase located in the intermembrane space. ΔΨ: membrane potential.

**Figure 3 cells-08-00286-f003:**
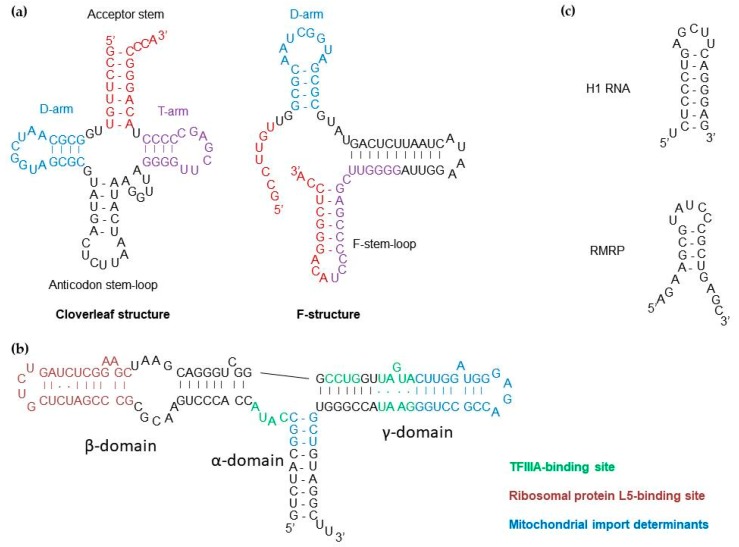
Import determinants of some nuclear-encoded RNAs localized in mitochondria. (**a**) Left, the cloverleaf structure of tRNA^Lys^_CUU_ (tRK1) of *S. cerevisiae*. Right, the F-structure of same RNA adopted upon binding with enolase. The 3′ part of the acceptor stem interacts with the 3′ side of the T-arm to form a new F-stem-loop which, together with the D-arm, functions as an import determinant. Adapted from [93]. (**b**) The secondary structure and major functional sites of human 5S rRNA. Interactions with the precursor of the mitochondrial ribosomal protein MRP-L18 and with the enzyme rhodanese (sites highlighted in blue) specify the 5S rRNA import into mitochondria. Adapted from [95]. (**c**) The stem-loop structures proposed to act as polynucleotide phosphorylase (PNPase)-binding import determinants in human H1 RNA and RMRP. Adapted from [19].

**Figure 4 cells-08-00286-f004:**
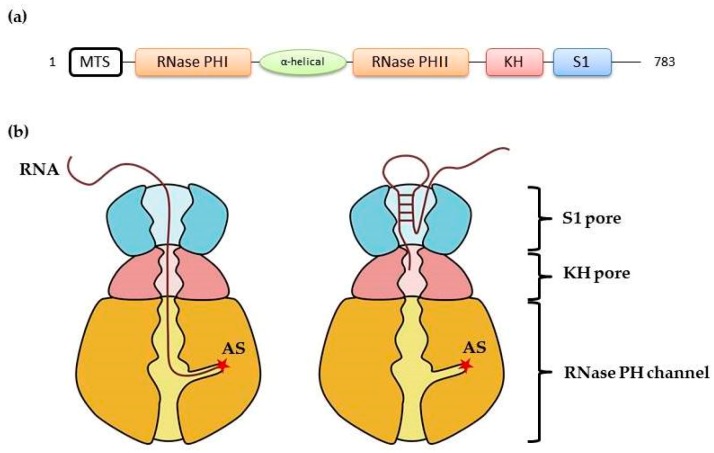
Human polynucleotide phosphorylase as an RNA import factor. (**a**) Domain organization of human PNPase. (**b**) Schematic model of a PNPase trimer bound to RNA. (Left) If the 3′ overhang is long enough, the bound RNA can access the active site (AS) and be degraded. (Right) Structured transcripts, e.g., imported small noncoding RNAs cannot enter the KH pore, which prevents their degradation [113,115].

**Figure 5 cells-08-00286-f005:**
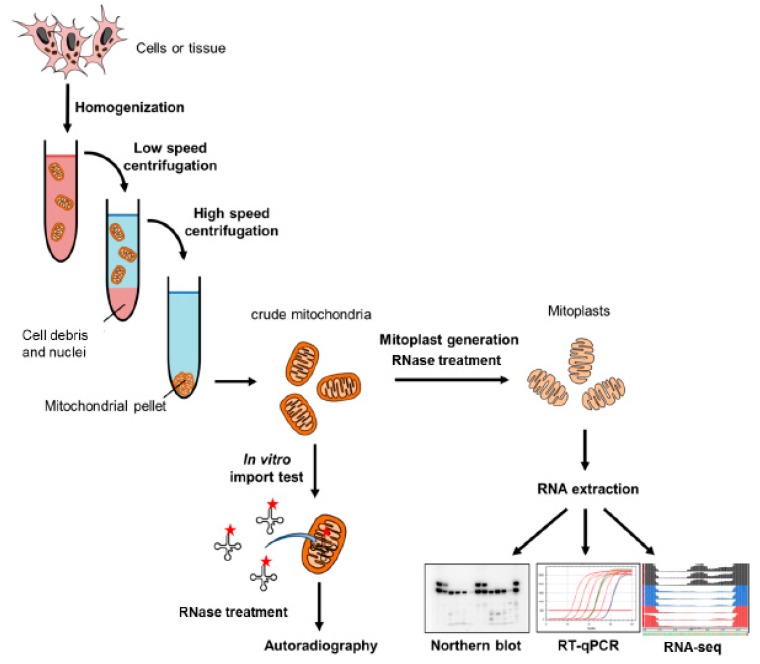
Analysis of mitochondrial RNA import by subcellular fractionation and in vitro import assay. Isolated mitochondria can be obtained by cell or tissue homogenization followed by differential centrifugation (as shown here) or antibody-mediated capture. Crude mitochondria can be used for in vitro import tests with radiolabeled transcripts or further purified to analyze endogenous mitochondria-localized RNAs.

**Figure 6 cells-08-00286-f006:**
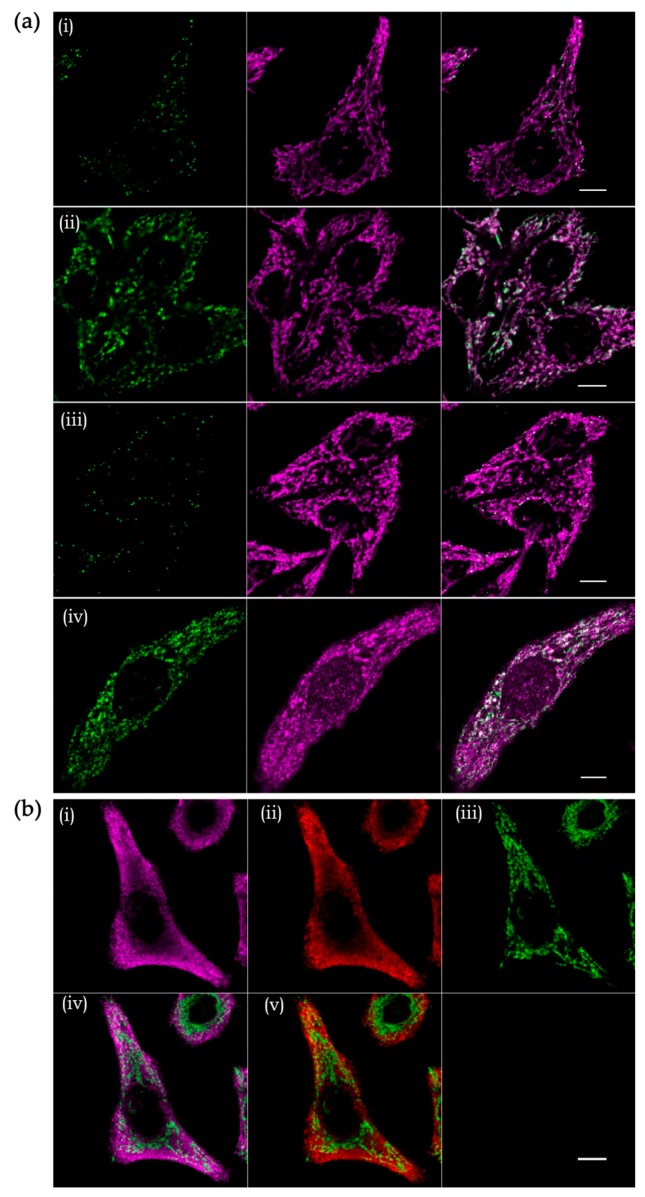
Detection of mitochondrial RNAs in HEK293T-Rex cells by RNA smFISH branched DNA and confocal microscopy. (**a**) mtDNA-derived transcripts: (i) *MT-CYTB* mRNA, (ii) 12S rRNA, (iii) mirror-12S rRNA, and (iv) tRNA^Val^. RNAs are shown in green (left column), and mitochondria stained with antibodies to TOM20 (i–iii) or mL38 (iv) are in magenta (middle column). The right column shows merged images. (**b**) Nuclear-encoded transcripts: (i) 5S rRNA and (ii) 5.8S rRNA. Panel (iii) shows TOM20. Panels (iv) and (v) are merged images for 5S rRNA/TOM20 and 5.8S rRNA/TOM20, respectively. Scale bars are 10 µm.

**Figure 7 cells-08-00286-f007:**
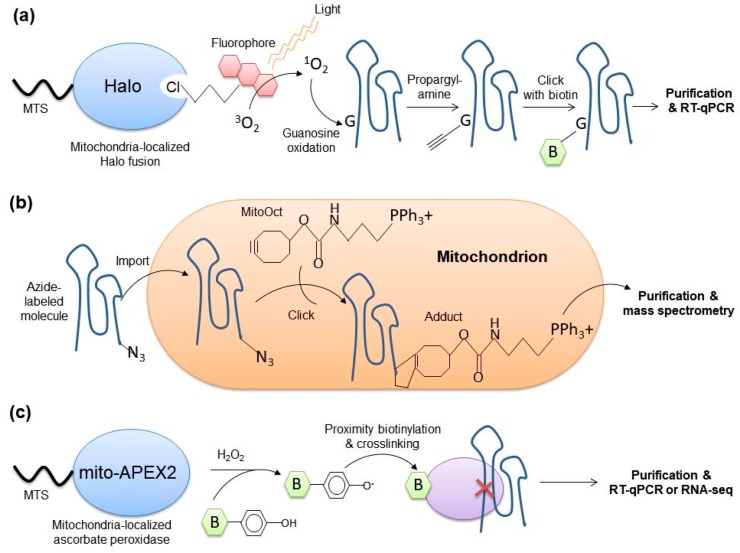
Detection of mitochondrial RNAs with spatially restricted tagging and crosslinking techniques: (**a**) spatially restricted guanosine oxidation [152,153], (**b**) ClickIn [154], (**c**) APEX-RIP [155] and Proximity-CLIP [156]. See main text for details.

**Table 1 cells-08-00286-t001:** Nuclear-encoded RNA species found in the human mitochondria.

RNA	Cytosolic Function	Evidence for Mitochondrial Localisation	Proposed Function in Mitochondria	References
Select tRNAs (including tRNA^Leu^_UAA_, tRNA^Gln^_UUG_, tRNA^Gln^_CUG_)	Translation	Deep sequencing of RNA isolated from mitoplasts RT-(q)PCR of RNA isolated from mitoplasts Enrichment in mitoplasts in comparison to crude mitochondria Import into isolated mitochondria	Mitochondrial translation under normal or stress conditions	[9,11]
5S rRNA	Component of the cytosolic ribosome	Deep sequencing of RNA isolated from mitoplasts RT-(q)PCR of RNA isolated from mitoplasts Northern blotting of RNA isolated from mitoplasts Enrichment in mitoplasts in comparison to crude mitochondria Import into isolated mitochondria Fluorescence microscopy (aptamer tagging, direct labeling)	Related to mitochondrial translation?	[11,12,13,14,15,16]
H1 RNA	Component of the nuclear RNase P required for pre-tRNA processing	Deep sequencing of RNA isolated from mitoplasts RT-(q)PCR of RNA isolated from mitoplasts Enrichment in mitoplasts in comparison to crude mitochondria Import into isolated mitochondria	Pre-tRNA processing?	[11,17,18,19]
RMRP	5.8S rRNA processing	Deep sequencing of RNA isolated from mitoplasts RT-(q)PCR of RNA isolated from mitoplasts Enrichment in mitoplasts in comparison to crude mitochondria Import into isolated mitochondria Electron microscopy	Mitochondrial RNA metabolism?	[11,19,20,21,22]
SAMMSON	Facilitates p32 targeting to the mitochondria in melanoma cells	RT-qPCR of RNA isolated from mitoplasts Fluorescence microscopy (FISH)	Unknown	[23,24]
hTERC	RNA component of telomerase	RT-PCR of RNA isolated from mitoplasts Import into isolated mitochondria	Mitochondria-cytosol communication	[25]
Various miRNAs (including miR-1, miR-181c, miR-378) and pre-miRNAs	Repression of mRNA translation	Deep sequencing of RNA isolated from mitoplasts RT-qPCR of RNA isolated from mitoplasts Enrichment in mitoplasts in comparison to crude mitochondria Fluorescence microscopy (FISH)	Repression or activation of mRNA translation, repression of transcription	[11,26,27,28,29,30,31,32,33,34]

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
