# Peer review of "Import of Non-Coding RNAs into Human Mitochondria: A Critical Review and Emerging Approaches"

_cells, 2019, doi:10.3390/cells8030286_

Round 1
Reviewer 1 Report
The manuscript by Jendard et al is a comprehensive review which discuss mechanisns of non-coding RNAs import into human mitochondria. Most importnatly it includes currently used methodologies and protocols which facilitate research in the field. This is a particulary interesting topic which deals with the nuclear-mitochondrial cross talk which includes an improntat dynamic newtowrk of molecules, among which, non-coding RNAs are the less well studied and understood. With the exception of mitochondrial tRNAs that contribute to severe clinical phenotypes in human, the functional and effective regulatory mitochondrial RNome is essentially unexplored in many levels. The informtaion in the current manuscript is assembled by experts in the field and the text is clearly presented and very well-written. I also found very useful the inclusion of experimental approaches which allow a better understanding of the exisiting knowledge which apperas updated in the refernec list. Overall, this is an excellent review which I highly recommend for publication and I feel that merits broader attention from researchers in the field. The authors prior to publication may consider some minor revisions to the following:
1. The sentence in lines 135-136 is a bit confusing, please rephrase or describe better.
2. In lines 328-336 the mitochondria isolation protocol with 1.0M and 1.5M sucrose step gradient solutions and ultracentrifugation in an Beckman SW28 or equivalent rotor, as appaears in “Clayton, D. A. & Shadel, G. S. Purification of Mitochondria by Sucrose Step Density Gradient Centrifugation. Cold Spring Harbor Protocols 2014, (2014)” could be also mentioned.
Author Response
The manuscript by Jendard et al is a comprehensive review which discuss mechanisns of non-coding RNAs import into human mitochondria. Most importnatly it includes currently used methodologies and protocols which facilitate research in the field. This is a particulary interesting topic which deals with the nuclear-mitochondrial cross talk which includes an improntat dynamic newtowrk of molecules, among which, non-coding RNAs are the less well studied and understood. With the exception of mitochondrial tRNAs that contribute to severe clinical phenotypes in human, the functional and effective regulatory mitochondrial RNome is essentially unexplored in many levels. The informtaion in the current manuscript is assembled by experts in the field and the text is clearly presented and very well-written. I also found very useful the inclusion of experimental approaches which allow a better understanding of the exisiting knowledge which apperas updated in the refernec list. Overall, this is an excellent review which I highly recommend for publication and I feel that merits broader attention from researchers in the field.
Response: We thank the referee for their enthusiasm regarding our manuscript.
The authors prior to publication may consider some minor revisions to the following:
1. The sentence in lines 135-136 is a bit confusing, please rephrase or describe better.
Response: We have rephrased the sentence to make it more informative and understandable.
2. In lines 328-336 the mitochondria isolation protocol with 1.0M and 1.5M sucrose step gradient solutions and ultracentrifugation in an Beckman SW28 or equivalent rotor, as appaears in “Clayton, D. A. & Shadel, G. S. Purification of Mitochondria by Sucrose Step Density Gradient Centrifugation. Cold Spring Harbor Protocols 2014, (2014)” could be also mentioned.
Response: We thank the referee for this suggestion. The corresponding citation (now [120]) has been added.
Reviewer 2 Report
Dear Editor,
the manuscript entitled "Import of non-coding RNAs into human mitochondria; a critical review and emerging research avenues" is very timely. And the authors, which are experts in the field, produced a very valuable resource to dive into the literature on this topic. I have only few minor comments:
-Abstract line 3: "selected non-coding RNA..."
-line 30-33: the sentence needs to be reworded
-line 39-42: I do not totally agree with the conclusion. There is no need for this molecules to act on mitochondrial gene expression (what does it means gene expression btw?) to be functional.
Also the english of the sentence could be improved.
-hTERC: The presence of hTERC in the mitochondria was shown in mitoplast purified with prot K, but not treated with RNAse!!
-line 99-100: reword as follow " For a long time, it was believed that the 5S rRNA was incorporated in the mitochondrial ribosome and needed for its function".
-all the manuscript change RT-PCR to RT-qPCR.
-Northern blot should be not capitalised (only Southern).
-Figure1: Not particularly fan of the design of the figure. In addition the right side is cut out
-Figure2: There should be lines separating the different stainings
-The title is not appealing..."emerging research avenues" should be "emerging technologies"?
Author Response
Dear Editor,
the manuscript entitled "Import of non-coding RNAs into human mitochondria; a critical review and emerging research avenues" is very timely. And the authors, which are experts in the field, produced a very valuable resource to dive into the literature on this topic.
Response: We thank the referee for their enthusiasm regarding our manuscript.
I have only few minor comments:
-Abstract line 3: "selected non-coding RNA..."
Response: Corrected.
-line 30-33: the sentence needs to be reworded
Response: We have rephrased the sentence to make it more understandable.
-line 39-42: I do not totally agree with the conclusion. There is no need for this molecules to act on mitochondrial gene expression (what does it means gene expression btw?) to be functional.
Also the english of the sentence could be improved.
Response: We agree with the referee and have modified the sentence in question accordingly.
-hTERC: The presence of hTERC in the mitochondria was shown in mitoplast purified with prot K, but not treated with RNAse!!
Response: The manuscript [25] is confusing in many respects; however, we believe to have correctly understood that the mitoplasts have been treated with micrococcal nuclease, as we presently note in the text.
-line 99-100: reword as follow " For a long time, it was believed that the 5S rRNA was incorporated in the mitochondrial ribosome and needed for its function".
Response: Done.
-all the manuscript change RT-PCR to RT-qPCR.
Response: We applied this change in all but two cases where RT-PCR has indeed been used in its non-quantitative version, those of tRNA(Gln) and hTERC.
-Northern blot should be not capitalised (only Southern).
Response: Corrected.
-Figure1: Not particularly fan of the design of the figure. In addition the right side is cut out
Response: There must have been a problem during the file compilation. The complete figure is uploaded separately.
-Figure2: There should be lines separating the different stainings
Response: We do not quite understand this comment. It might be that the referee meant Figure 6. In the case our guess is correct, we have introduced the separators between the panels.
-The title is not appealing..."emerging research avenues" should be "emerging technologies"?
Response: We have modified the title in a similar vein to read “emerging approaches”.
Reviewer 3 Report
In the current manuscript the authors reviewed certain aspects on the import of non-coding RNAs into human mitochondria.
Overall, the review is timely and written in a clear manner. It is appreciated that the authors also discuss methods and methodological aspects which are importnat to help to see limitations of the studies so far and to show emerging new avenues.
In addition, I have several aspects which should be considered by the authors to make the review even more clear.
Since the authors emphasize in their title the "human" mitochondrial aspect they should clearly indicate throughout their paper which studies stem from yeast, funghi, plants etc and which are from mammalian/human system. At current this is very difficult and needs to be included.
The discussion on RNaseP (also in table1) should contain information about the difference between human and yeast; in yeast it is encoded in mitochondrial DNA and in humans nuclear. Further, mammalian RNaseP does not have a RNA subunit. Please elaborate on this.
The discussion on nuclear and mitochondrial tRNA, in particular that of tRNA-Gln and tRNA-Glu should be expanded and the differences due to metabolic conversions should be more emphasized
encoded
Author Response
Comments and Suggestions for Authors
In the current manuscript the authors reviewed certain aspects on the import of non-coding RNAs into human mitochondria.
Overall, the review is timely and written in a clear manner. It is appreciated that the authors also discuss methods and methodological aspects which are importnat to help to see limitations of the studies so far and to show emerging new avenues.
Response: We thank the referee for their enthusiasm regarding our manuscript.
In addition, I have several aspects which should be considered by the authors to make the review even more clear.
Since the authors emphasize in their title the "human" mitochondrial aspect they should clearly indicate throughout their paper which studies stem from yeast, funghi, plants etc and which are from mammalian/human system. At current this is very difficult and needs to be included.
Response: We fully agree with the referee; it is particularly important for the miRNA studies made on various mammalian species. We have specified the biological system whenever possible.
The discussion on RNaseP (also in table1) should contain information about the difference between human and yeast; in yeast it is encoded in mitochondrial DNA and in humans nuclear. Further, mammalian RNaseP does not have a RNA subunit. Please elaborate on this.
Response: We thank the referee for this suggestion. We have expanded the text accordingly.
The discussion on nuclear and mitochondrial tRNA, in particular that of tRNA-Gln and tRNA-Glu should be expanded and the differences due to metabolic conversions should be more emphasized
Response: We thank the referee again for this useful suggestion. We included this information and the accompanying references in the corresponding section.